# Phenotypic Diversity in Pre- and Post-Attachment Resistance to *Striga hermonthica* in a Core Collection of Rice Germplasms

**DOI:** 10.3390/plants12010019

**Published:** 2022-12-20

**Authors:** Hiroaki Samejima, Yukihiro Sugimoto

**Affiliations:** Graduate School of Agricultural Science, Kobe University, 1-1 Rokkodai, Nada, Kobe 657-8501, Japan

**Keywords:** rice, *Oryza sativa*, *Striga hermonthica*, phenotyping, germination-inducing activity, rhizotron, multiple resistance

## Abstract

In sub-Saharan Africa, upland rice cultivation is expanding into rainfed areas endemic to the root parasitic weed *Striga hermonthica*. We evaluated the *Striga* resistance of 69 accessions from the World Rice Core Collection (WRC) to estimate the phenotypic diversity within the *Oryza sativa* species. Pre-attachment resistance was screened based on the germination-inducing activities of the root exudates, while post-attachment resistance was screened through rhizotron evaluation. The 69 WRC accessions showed a wide variation in both pre- and post-attachment resistance. Root exudates of one accession induced 0.04% germination, and those of some accessions displayed >80% germination. In the evaluation of post-attachment resistance, the successful parasitism percentages ranged from 1.3% to 60.7%. The results of these resistance evaluations were subjected to cluster analysis, which recognized five groups: group I of 27 accessions, with high pre- and post-attachment resistance; group II of 12 accessions, with high post-attachment resistance but moderate pre-attachment resistance; group III of 4 accessions, with low pre-attachment resistance; group IV of 13 accessions, with low post-attachment resistance; and group V of 13 accessions, with low pre- and post-attachment resistance. The wide variation found in the WRC accessions will help to elucidate the genetic factors underpinning pre- and post-attachment resistance.

## 1. Introduction

Rice (*Oryza sativa* L.) is the staple food for over half of the world’s population and is of increasing nutritional importance for large portions of the population in Africa [1]. In sub-Saharan Africa, rice is the third-largest source of food energy, with a rising demand due to population growth and changes in eating habits [2]. The expansion of rice cultivation by converting fallow, maize, and sorghum fields into rainfed rice fields is one method that will help to close the gap between rice demand and production in the region [3]; however, these rainfed areas are sometimes infested by the root parasitic weed *Striga hermonthica* (Delile) Benth. The agricultural damage caused by *Striga* threatens food security for the growing population in sub-Saharan Africa. The Food and Agricultural Organization of the United Nations estimates that, across the continent, *Striga* causes annual losses exceeding US$7 billion, adversely affecting more than 300 million people [4]. Thus, the increased production of rice in rainfed areas is likely to be associated with increased *S. hermonthica* problems. Under the prevailing farming conditions in sub-Saharan Africa, resistant crop varieties have been proposed as the most integral, cost effective, and easy to adopt component of an integrated *Striga* management strategy [4].

Many *Striga*-resistant rice varieties that have adapted to the growing environments in Africa have already been identified [3]; however, an increased variation in *Striga*-resistant rice germplasms could provide new practical rice varieties for use in areas of *Striga* infestation. Large collections of rice germplasms have been conserved in gene banks or germplasm centers around the world [5]. Core collections are small collections of germplasms that represent the genetic diversity in the larger gene bank collections, and are a powerful tool that enables the efficient use of genetic resources [6]. The rice accessions contained in these core collections, however, have not yet been evaluated for *Striga* resistance.

The lifecycle of parasitic plants can be divided into the pre-attachment, free living phase, and the post-attachment, parasitic phase. Similarly, crop resistance to parasitic plants can be defined as pre-attachment and post-attachment [7]. As obligate hemiparasites, *Striga* seeds germinate only when exposed to host root-derived stimulants [8]. Pre-attachment resistance occurs when host plants prevent parasite attachment by producing lower amounts or less effective types of germination stimulants [9,10]. The germination stimulants are collectively known as strigolactones. They function as rhizosphere signaling molecules to induce germination of root parasitic weed seeds. The strigolactones also promote the formation of mutually beneficial symbioses between plants and arbuscular mycorrhizal fungi [11], which provide phosphate to plants. In addition, these molecules function as phytohormones, regulating various aspects of plant architecture and development [12,13]. In contrast, post-attachment resistance prevents parasite development [7]. For example, the lowland rice cultivar Nipponbare has been reported to exhibit a post-attachment resistance that is associated with an incompatibility of its cortex, endodermis, and root stele to *Striga* [14,15]. Specifically, site-specific lignification of the endodermis at the *Striga* infection site is crucial to Nipponbare’s *Striga* resistance, as indicated by its increased susceptibility to *Striga* when its S-, G-, and H-unit lignin balance is disrupted [16]. Several rice varieties have been reported as resistant to *Striga* at the pre- and post-attachment phases [3,17].

In this study, we evaluated the pre- and post-attachment resistance in rice accessions found in a core collection, which is assumed to represent the genetic diversity of >32,000 accessions [18]. The evaluation was conducted under controlled conditions in a laboratory setting to lessen any confounding environmental factors. We collected and measured the germination-inducing activity of root exudates from a hydroponic solution to determine pre-attachment resistance. Post-attachment resistance was determined by observing the growth of *Striga* seedlings that were artificially attached to rice roots in a rhizotron. We estimated the phenotypic diversity within the *Oryza sativa* species using these two indices.

## 2. Results

### 2.1. Germination Test to Phenotype for Pre-Attachment Resistance

The root exudates from the 69 accessions displayed a 0.04–81.0% germination rate, with significant (*p* < 0.05) differences found between the accessions (Figure 1). Nipponbare, which is known for its *Striga* pre-attachment resistance, exhibited low germination-inducing activities (0.1%), or in other words, displayed a high pre-attachment resistance (Figure 1). Kasalath induced germination at a rate of 64.4% (Figure 1). The frequency distribution for pre-attachment resistance showed a bimodal distribution (Figure 2).

### 2.2. Rhizotron Evaluation to Phenotype for Post-Attachment Resistance

Successful parasitism varied widely among the 69 accessions and represented 1.3–60.7% of the original inoculums with significant (*p* < 0.05) differences between the accessions (Figure 3). The proportion of successful parasitism were relatively low (16.0%) in Nipponbare, a known *Striga* post-attachment resistant variety (Figure 3). Kasalath retained 5.3% of the original inoculums (Figure 3). The frequency distribution for post-attachment resistance approximately fitted a normal distribution (Figure 4).

### 2.3. Clustering of the Accessions Based on Pre- and Post-Attachment Resistance

The cluster analysis divided the 69 accessions into five groups (Figure 5). Figure 6 shows the pre- and post-attachment resistance of each accession in the five groups. Group I consisted of 27 accessions that displayed a high resistance during both the pre- and post-attachment phases. Nipponbare was in this group. Group II consisted of 12 accessions that displayed an intermediate pre-attachment resistance and a high post-attachment resistance, comparable to group I. Group III consisted of 4 accessions that displayed a low pre-attachment resistance and a high post-attachment resistance, comparable to or higher than those of group I. Group IV consisted of 13 accessions that displayed high pre-attachment resistance that was comparable to that of group I, and a low post-attachment resistance. Group V consisted of 13 accessions that displayed a low resistance at both the pre- and post-attachment phases. The number and proportion of accessions allocated to each group by type of rice (i.e., japonica, indica I, and indica II) are provided in Table 1. All groups included each rice type, except for group III, which lacked the indica II type. The highest proportion of japonica and indica II accessions were in group I with 35.7% and 51.5%, respectively. For the indica I type, 22.7% were in group I, but more (36.4%) were in group V. Three of the 4 accessions in group III were indica I type, which included Kasalath. 

## 3. Discussion

### 3.1. Differences among Accessions in Pre- and Post-Attachment Resistance

We focused on Nipponbare and Kasalath, whose pre- and post-attachment resistance have been previously reported, to evaluate the reliability of the phenotyping methods used in this study. The results in this study were in line with those reported by Samejima et al. [17], in which Nipponbare had a high pre- and post-attachment resistance and Kasalath had a high post-attachment resistance to *Striga* from sorghum fields in Sudan. Gurney et al. [14] reported that Kasalath had low post-attachment resistance to *Striga* from maize fields in Kenya. The different Kasalath evaluation results are probably due to the ecotype differences of the *Striga* seedlings used for each study. Post-attachment resistance to *Striga* was higher in Nipponbare than in Kasalath when *Striga* was collected from maize fields in Kenya, but was comparable when the *Striga* was collected from rice fields in the Ivory Coast, and was lower in Nipponbare than in Kasalath when *Striga* was collected from millet fields in Gambia [19]. In addition, Bevawi [20] reported that the ranking of sorghum cultivars varied depending on *Striga* ecotypes, with respect to germination-inducing activities of their root exudates. Germination-inducing activities of several synthetic strigolactones were higher for *Striga* from Sudan than those from Burkina Faso, Niger, and Kenya [21]. These findings suggest that results for pre- and post-attachment resistance can vary depending on *Striga* ecotypes. We judged, however, the phenotyping methods used in our study to be reliable for both pre- and post-attachment resistance to *Striga* collected from sorghum fields in Sudan.

In this study, the pre- and post-attachment resistance varied widely among the 69 WRC accessions. Understanding the genetic nature of these variations requires the integration of genomics and phenotyping. A genome-wide association study (GWAS) is a powerful and efficient tool to analyze the natural allelic variation associated with agronomic traits, using a diverse population of accessions [6]. For example, GWAS with 173 and 206 diverse sorghum genotypes determined the genetic factors that underpin pre- and post-attachment resistance, respectively [22,23]. Although the WRC is a very small population categorized as a mini-core collection [5,24], it can be an effective population for detecting the genes responsible for *Striga* resistance by GWAS, if there are strong quantitative trait loci (QTLs) associated with the trait of interest and if the number of related QTLs is relatively small [6]. GWAS detected *waxy* and *GS3* as genes associated with seed amylose content and grain length, whose distribution was bimodal and approximately normal, respectively [6]. Thus, the detection of the genes associated with the desired phenotypic characteristics may be possible using GWAS on the WRC accessions, as indicated by the bimodal and normal distributions for the pre- and post-attachment resistance observed in this study, respectively. 

Elucidating the physiological mechanisms for the observed resistances in each accession is a topic for future study. A compositional analysis of root exudates is necessary to reveal the mechanisms for pre-attachment resistance, which includes absent or reduced germination stimulants production and exudation of germination inhibitors [25]. After *Striga* germination, reduction of haustorium formation and mechanical barriers by thickening of host root cell-walls are candidate resistance mechanisms [25]. Furthermore, once *Striga* starts penetrating the host root tissues and tries to connect to the host vascular system, different post-attachment resistance mechanisms might be included, for example, the synthesis or release of cytotoxic compounds in the form of phenolic acids or phytoalexins within infected host root cells, the formation of physical barriers to prevent possible pathogen ingress and growth, and programmed cell death in the form of a hypersensitive response at the point of parasite attachment [25].

### 3.2. Combination of Pre- and Post-Attachment Resistance in Each Accession

Rice varieties with multiple resistances (i.e., high resistance at both the pre- and post-attachment phases) will be useful genetic resources for rice cultivation in areas where *Striga* is endemic. When the 69 WRC accessions were clustered into the five groups, the largest number of accessions were categorized into group I (high resistance at both pre- and post-attachment phases). To develop the WRC, the RFLP data on 332 accessions were subjected to cluster analysis and 67 groups were recognized, then a single accession from each of the 67 groups was selected [18]. The findings in our study suggest that many rice varieties not included in the WRC may have the same multiple resistance. Several rice varieties have thus far been reported as possessing this multiple resistance, such as Nipponbare, Umgar [17], CG14, and NERICA 1 [3]. The accessions with only one resistance, such as those in groups II (high post-attachment resistance), III (high post-attachment resistance), and IV (high pre-attachment resistance), could also be good genetic resources for improving *Striga* resistance in rice. In contrast, the accessions in group V (low resistance at both pre- and post-attachment phases) suggest that many rice varieties not included in the WRC may be very susceptible to *Striga*. Careful attention should be paid to the responses of different rice varieties to *Striga*, before their introduction to areas under *Striga* infestation. 

The distribution percentages of the japonica, indica I, and indica II types in the five groups seemed to vary. Wide variation in the pre- and post-attachment resistance within each of these types was found, as indicated by the inclusion of all three types in each group (except for a lack of indica II in group III). Japonica and indica II tended to be in group I (high resistance at both pre- and post-attachment phases), while indica I tended to be in group V (low resistance at both pre- and post-attachment phases). In addition, three of the four group III accessions (low pre-attachment resistance and high post-attachment resistance) were of the indica I type. It should be noted that how accurately the WRC accessions represent each type is unclear. Further study is required to understand the differences in *Striga* resistance among the indica I, indica II, and japonica types.

## 4. Materials and Methods

### 4.1. Plant Materials

The 69 rice accessions found in the National Agriculture and Food Research Organization (NARO) World Rice Core Collection (WRC) were used to evaluate pre- and post-attachment resistance to *Striga*. The 69 accessions included 14 japonica, 22 indica I, and 33 indica II types that were originally collected from 19 different countries [26]. Sixty-seven of these accessions retained 91% of the 554 RFLP alleles in 332 accessions, selected based on passport data from the entire NARO gene bank collection [18]. The WRC consists of these 67 accessions and 2 reference accessions for japonica (Nipponbare) and indica (Kasalath) discrimination in the RFLP survey [18]. *S. hermonthica* seeds were collected from sorghum fields in Sudan.

### 4.2. Germination-Inducing Activity of Root Exudates

The germination bioassay followed the protocol described by Samejima et al. [17]. The rice accession seeds were sown on moist filter paper in a Petri dish and incubated in the dark at 30 °C for 4 days. Next, each rice seedling was transplanted into a 50 mL plastic tube that was wrapped in aluminum foil to block-out any light. The plants were then hydroponically grown in a 40% Long Ashton solution in a growth chamber at 30 °C, with a 12 h photoperiod using fluorescent lights (photosynthetic active radiation of 220 μmol m^−2^ s^−1^) until 28 days after sowing (DAS). Roots were thoroughly rinsed on 29 and 30 DAS with fresh tap water, and then grown in fresh tap water for two days. After two days, the water lost from each tube was replenished. On 31 DAS, 10 mL of the aqueous solution containing the root exudates was extracted with 2 mL of ethyl acetate, as reported previously [27]. After centrifuging for 5 min at 3500 rpm, 40 μL of the ethyl acetate extract was applied to a new 8 mm diameter glass fiber disk. The disks were dried at room temperature for 2 h. Each treated disk was overlaid with an 8 mm diameter glass fiver disk containing conditioned *Striga* seeds moistened with 40 μL distilled water. The *Striga* seeds had been conditioned under sterilized conditions for 12 days on the disks (approximately 50 seeds per disk). The disks containing the conditioned *Striga* seeds were tapped on filter paper to remove any excess water, before being placed on the disks treated with the root exudates. *Striga* germination was examined using a stereomicroscope after incubation in the dark at 30 °C for 1 day. The synthetic strigolactone GR24 was used at 0.34 μM as a positive control and tap water was used as a negative control, which induced 54.1–89.2% and 0.0–3.1% germination, respectively. Three plants per accession were used, and the root exudates from each plant were tested in three technical replicates. 

### 4.3. Rhizotron Experiment

The rhizotron experiment followed the protocol described by Samejima et al. [17]. The rice accession seedlings were prepared in the same manner as in the *Striga* germination test. Each seedling was transferred to a 10 mL test tube and grown hydroponically in a 40% Long Ashton solution for 6 days, in a growth chamber at 30 °C with a 12 h photoperiod using fluorescent lights (photosynthetic active radiation of 220 μmol m^−2^ s^−1^). Next, a 10-day-old seedling of each accession (with five replicates) were transferred to a rhizotron comprising of a 15 cm Petri dish filled with rock wool that was overlaid by glass fiber filter paper that was watered with 40% Long Ashton solution. The seedlings were then grown in the same growth chamber for an additional 10 days. The *Striga* seeds were surface sterilized by immersion in 0.75% (*w*/*v*) NaClO that contained a few drops of Tween 20, and then were subjected to sonication for 3 min in an ultrasonic cleaner. After rinsing with distilled water, the seeds were dried in a laminar hood, then were pre-treated (conditioned) for 12 days on distilled water-saturated filter paper in a Petri dish. The conditioned *Striga* seeds were then treated in separate Petri-dishes with GR24 at 0.34 μM and incubated in the dark at 30 °C for 1 day prior to inoculation. The roots of each 20-day-old rice seedling were inoculated with 30 pre-germinated *Striga* seeds, carefully picked up with a pair of tweezers from the Petri dish, and subsequently incubated in the same growth chamber. *Striga* seedling growth was observed 21 days after inoculation using a stereomicroscope. The number of successful parasitism was counted.

### 4.4. Statistical Analysis

Mean comparison between the accessions was determined for both the pre- and post-attachment resistance using JMP software (version 16.1.0; SAS Institute Inc. Cary, NC, USA). The germination percentage values as an index of pre-attachment resistance were transformed to arcsine, then compared by Tukey’s HSD test and back-transformed [28]. For the rhizotron evaluation of post-attachment resistance, the number of successful parasitisms were examined by Tukey’s HSD Test, since all rice seedlings were inoculated with 30 pre-germinated *Striga* seeds. Cluster analysis was performed to divide the accessions into groups using the statistical package R (version 3.4.0; R Development Core Team, 2017) with the built-in *hclust* function, using the complete method and Euclidean distance.

## 5. Conclusions

In this study, we phenotyped 69 WRC accessions for pre- and post-attachment resistance to *Striga*. The wide diversity in the WRC observed here may help to elucidate the genetic factors that underpin pre- and post-attachment resistance, since the genetic diversity found in the WRC accessions has been well characterized by whole genome sequencing [6]. The existence of accessions displaying only one type of resistance, such as those in groups II, III, and IV, suggests that pre- and post-attachment resistance is genetically independent. The accessions in groups I–IV could provide good genetic resources for improving *S. hermonthica* resistance in rice. The presence of accessions in group V indicates the continued need to carefully consider *Striga* resistance when expanding rice cultivation in Africa.

## Figures and Tables

**Figure 1 plants-12-00019-f001:**
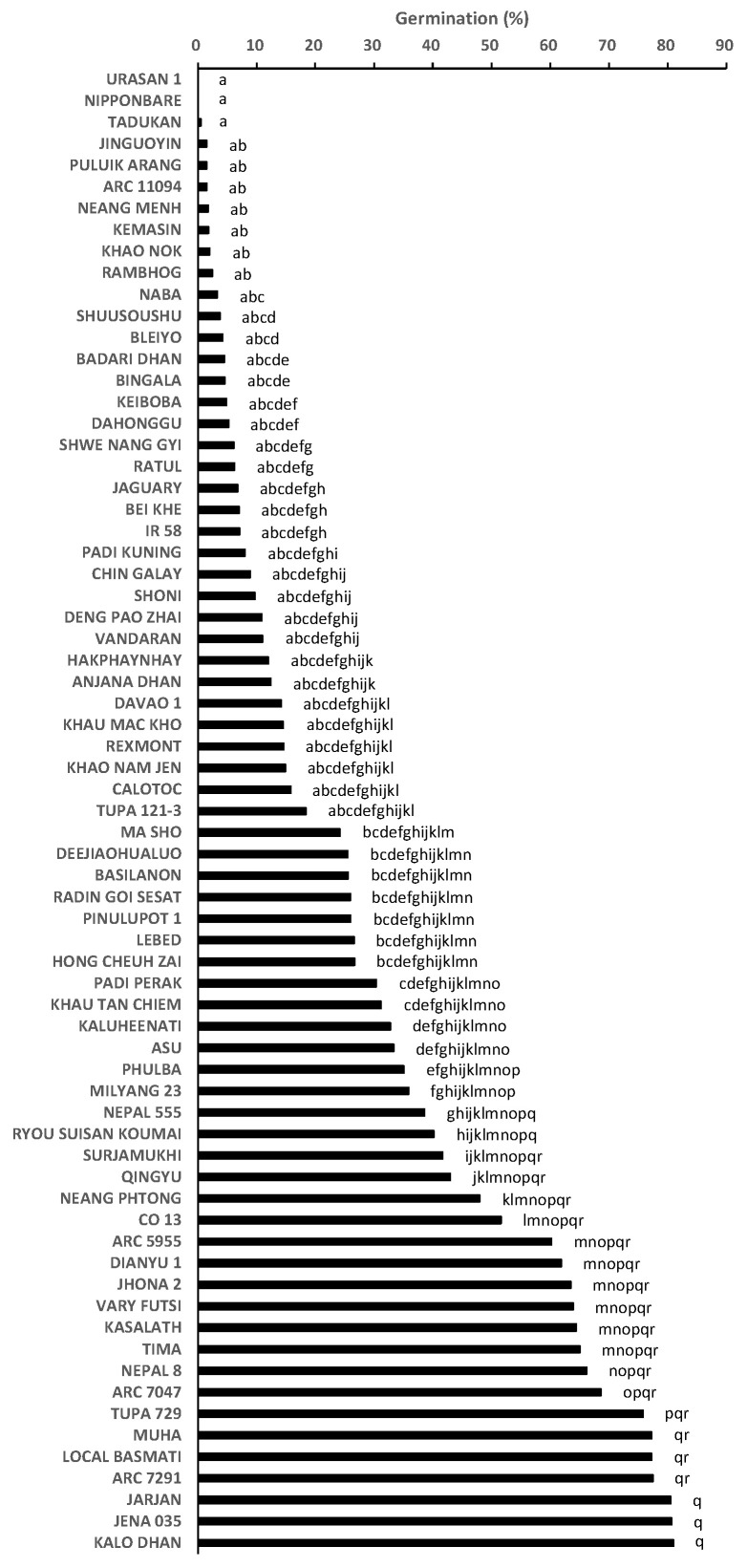
Germination-inducing activity of root exudates (pre-attachment resistance) of 69 accessions in World Rice Core Collection. The percentage values were arcsine transformed, compared by multiple mean-separation tests among the accessions and back-transformed. Means followed by a common letter are not significantly different by Tukey’s HSD test (*p* < 0.05). The synthetic strigolactone GR24 was used at 0.34 μM as a positive control, and tap water was used as a negative control, which induced 54.1–89.2% and 0.0–3.1% germination, respectively.

**Figure 2 plants-12-00019-f002:**
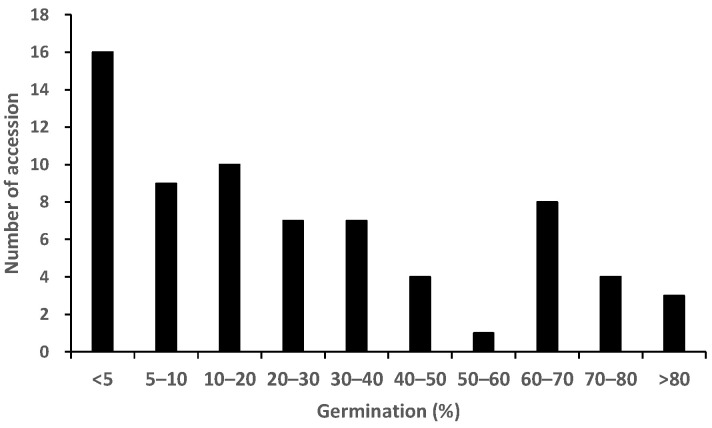
Frequency distribution of 69 accessions in World Rice Core Collection for percentage of germination-inducing activity of root exudates (pre-attachment resistance).

**Figure 3 plants-12-00019-f003:**
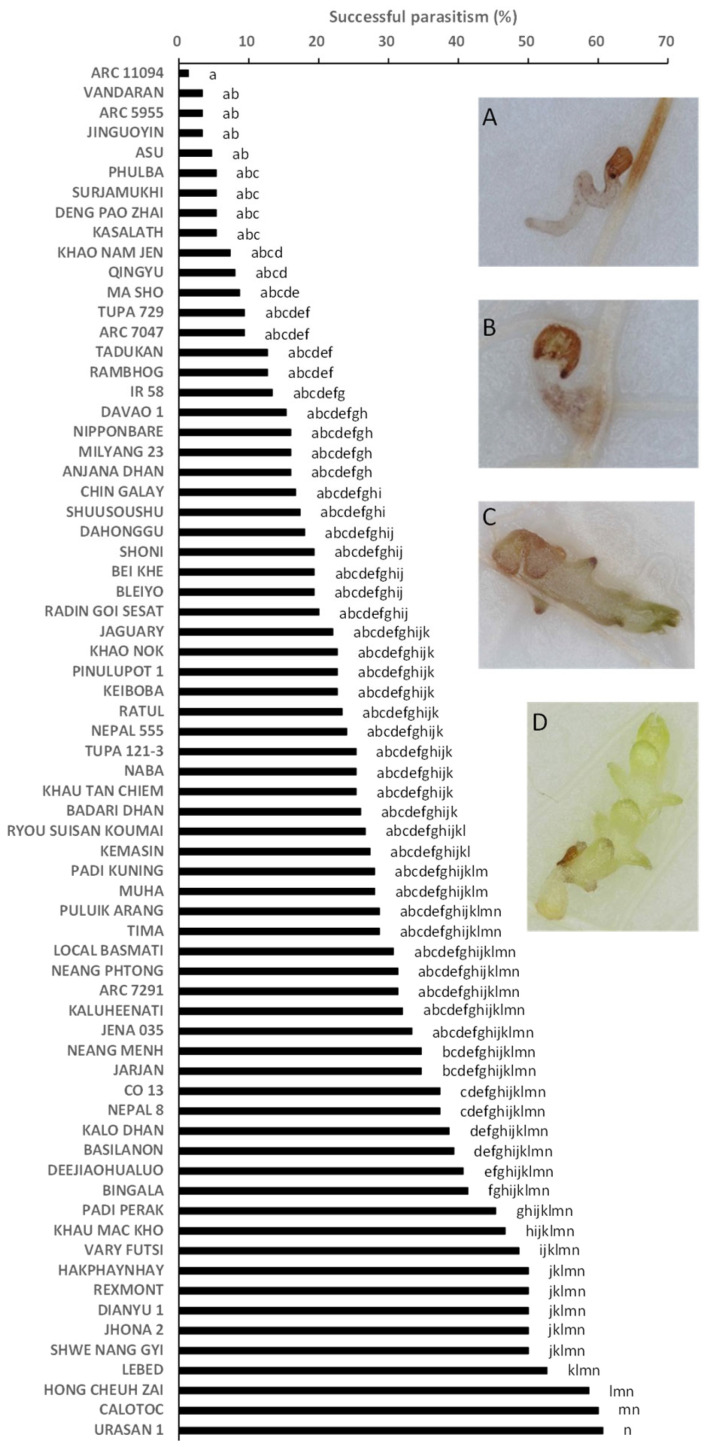
Percentage of successful parasitism out of 30 seedlings of *S. hermonthica* (post-attachment resistance) on the root of 69 accessions in World Rice Core Collection. Means followed by a common letter are not significantly different by Tukey’s HSD test (*p* < 0.05). Insert **A**, **B**, **C** is unsuccessful parasitism because of failure to attach, no shoot emergence, and withering, respectively. Insert **D** is successful parasitism.

**Figure 4 plants-12-00019-f004:**
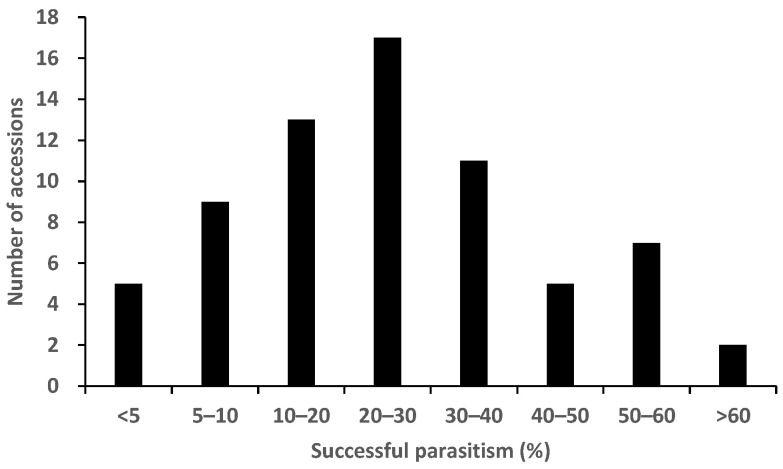
Frequency distribution of 69 accessions in World Rice Core Collection for percentage of successful parasitism (post-attachment resistance).

**Figure 5 plants-12-00019-f005:**
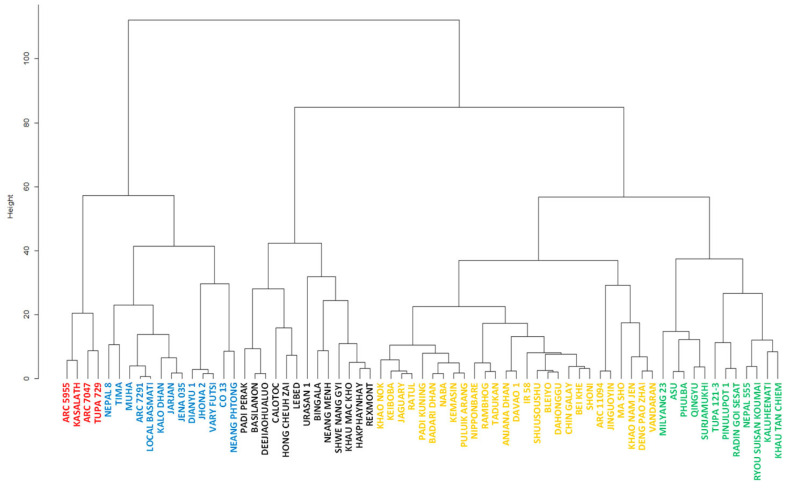
Cluster dendrogram for the classification of 69 accessions in World Rice Core Collection based on pre- and post-attachment resistance.

**Figure 6 plants-12-00019-f006:**
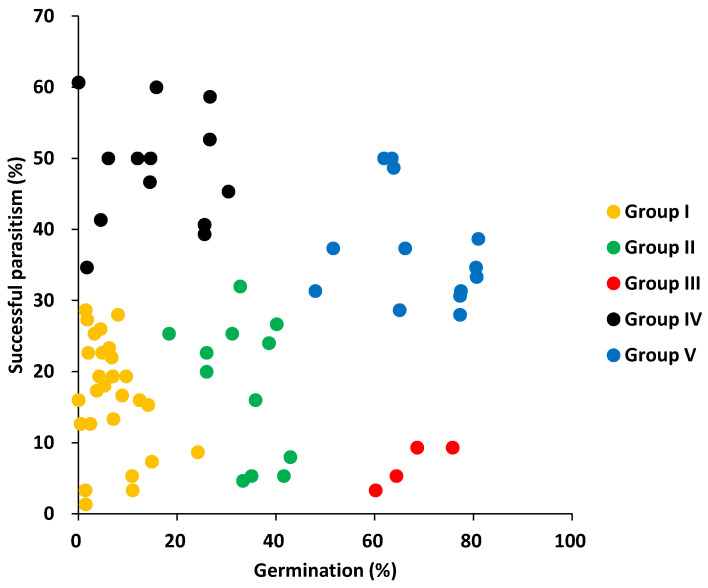
Germination-inducing activity (pre-attachment resistance) and percentage of successful parasitism (post-attachment resistance) of each accession in five groups.

**Table 1 plants-12-00019-t001:** Number and proportion of accessions allocated to each group within japonica, indica I, and indica II type.

Type	Group I	Group II	Group III	Group IV	Group V	Sum
japonica	5 (35.7)	2 (14.3)	1 (7.1)	4 (28.6)	2 (14.3)	14 (100)
indica I	5 (22.7)	4 (18.2)	3 (13.6)	2 (9.1)	8 (36.4)	22 (100)
indica II	17 (51.5)	6 (18.2)	0 (0.0)	7 (21.2)	3 (9.1)	33 (100)

## Data Availability

Data are available from the authors upon request.

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
