# Peer review of "Phenotypic Diversity in Pre- and Post-Attachment Resistance to *Striga hermonthica* in a Core Collection of Rice Germplasms"

_plants, 2022, doi:10.3390/plants12010019_

Round 1
Reviewer 1 Report
Dear authors,
The present study „Phenotypic diversity in pre- and post-attachment resistance to Striga hermonthica in a core collection of rice germplasms“ is very interesting, and thoroughly prepared and also very important for the scientific community as well as for the applied science concerning agriculture crop and its importance for present and future food security.
The experimental design and its statistical evaluation seems to be suitable. For material and methods: it is already mentioned that the test-series included positive and negative controls (with water only) for germination - the tested values are already mentioned in this chapter – but they should also be displayed as part of the results in terms of .
For the experiments images of Striga germination on rice-exudates in different stages as well as imaging of rice-seedlings in ccontact with Striga with successful and unsuccessful parasitism events would be favourable for this manuscript.
Author Response
Thank you for providing importnd suggestion.
Point 1: For material and methods: it is already mentioned that the test-series included positive and negative controls (with water only) for germination - the tested values are already mentioned in this chapter – but they should also be displayed as part of the results in terms of .
Response 1: We agree with you and have added the gergermination rates of the positve and negative controls in the legend of Figure 1.
Point 2: For the experiments images of Striga germination, rice-exudates in different stages would be favourable for this manuscript.
Response 2: We have data only for 30-day-old rice plants. As you suggests, germination-inducing activities in rice root exudates may fluctuate with its growth. However, we believe that differences of the activities exhibited by 30-day-old rice plants are good indication to compare germination-induction potential of the varieties. In our previous study (reference 17), the peaks of germination-inducing activity of rice varietes were obseved with around 30-day-old rice plants.
Point 3: For imaging of rice-seedlings in contact with Striga, successful and unsuccessful parasitism events would be favourable for this manuscript.
Response 3: We agree with you that images of successful and unsuccessful parasitism events would help the readers to understand the results of rhizotoron experiment. We have added these images as inserts of Figure 1.
Reviewer 2 Report
This is a very nice study and provides valuable phenotypic data that could be used with genomic sequencing to determine genetic control of Striga resistance in rice. It is a valuable contribution to an understudied problem of rice cultivation in Africa.
One very minor text improvement would be to revise the sentence running from line 56-61. It is a bit long and could perhaps be divided in two. At least insert a comma after "stimulants" and again after "strigolactones" on line 57.
I know these phenotyping protocols are not easy to do or interpret, but the way you presented the results was clear. Congratulations!
Author Response
Thank you for providing importnd suggestion.
Point 1: One very minor text improvement would be to revise the sentence running from line 56-61. It is a bit long and could perhaps be divided in two. At least insert a comma after "stimulants" and again after "strigolactones" on line 57.
Resoponse 1: We agree with you that the sentence was too long and difficult to understand. We have divieded it into three sentences (Line 58-63).